# The Protective Effects of Mcl-1 on Mitochondrial Damage and Oxidative Stress in Imiquimod-Induced Cancer Cell Death

**DOI:** 10.3390/cancers16173060

**Published:** 2024-09-02

**Authors:** Shu-Hao Chang, Kai-Cheng Chuang, Zheng-Yi Li, Mao-Chia Chang, Kuang-Ting Liu, Chien-Sheng Hsu, Shi-Wei Huang, Mu-Chi Chung, Shih-Chung Wang, Yi-Ju Chen, Jeng-Jer Shieh

**Affiliations:** 1Institute of Biomedical Sciences, National Chung Hsing University, Taichung 402202, Taiwan; shchang0501@dragon.nchu.edu.tw (S.-H.C.); d108059001@mail.uchu.edu.tw (Z.-Y.L.); g112059008@mail.nchu.edu.tw (M.-C.C.); shaintane@aftygh.gov.tw (K.-T.L.); 164997@cch.org.tw (C.-S.H.); 2Department of Life Sciences, National Chung Hsing University, Taichung 402202, Taiwan; ckchuang@dragon.nchu.edu.tw; 3Department of Pathology & Laboratory Medicine, Taoyuan Armed Forces General Hospital, Taoyuan 325208, Taiwan; 4Frontier Molecular Medical Research Center in Children, Changhua Christian Children Hospital, Changhua 500209, Taiwan; 5Center for Cell Therapy and Translation Research, China Medical University Hospital, Taichung 404327, Taiwan; 6Division of Nephrology, Department of Medicine, Taichung Veterans General Hospital, Taichung 40705, Taiwan; 7PhD Program in Translational Medicine, National Chung Hsing University, Taichung 402202, Taiwan; 8Department of Biotechnology, Asia University, Taichung 413305, Taiwan; 9Rong Hsing Research Center for Translational Medicine, National Chung Hsing University, Taichung 402202, Taiwan; 10Division of Pediatric Hematology/Oncology, Changhua Christian Children Hospital, Changhua 500209, Taiwan; 68122@cch.org.tw; 11Department of Post-Baccalaureate Medicine, College of Medicine, National Chung Hsing University, Taichung 402202, Taiwan; yjchenmd@mail.vghtc.gov.tw; 12Department of Dermatology, Taichung Veterans General Hospital, Taichung 40705, Taiwan; 13Department of Education and Research, Taichung Veterans General Hospital, Taichung 40705, Taiwan

**Keywords:** imiquimod, Mcl-1, ROS, mitochondrial dynamics, mitophagy

## Abstract

**Simple Summary:**

Imiquimod (IMQ) is clinically used in the treatment of various skin malignancies. We previously showed that IMQ-induced apoptosis and autophagic cell death in skin cancer cells are ROS-dependent. Additionally, IMQ-induced apoptosis is associated with a decrease in Mcl-1 levels. However, the exact role of Mcl-1 in IMQ-induced apoptosis, including its protective mechanisms and physiological function in cancer cells, remains unclear. This study demonstrated that the overexpression of Mcl-1 or IL-6-induced Mcl-1 upregulation reversed mitochondrial dysfunction, mitochondrial fission, and mitophagy in IMQ-treated cancer cells and protected them from IMQ-induced apoptosis. These results provide significant insights supporting the role of Mcl-1 in mitochondria and suggest that it may be a potential target for cancer research and therapy.

**Abstract:**

Mitochondria, vital organelles that generate ATP, determine cell fate. Dysfunctional and damaged mitochondria are fragmented and removed through mitophagy, a mitochondrial quality control mechanism. The FDA-approved drug IMQ, a synthetic agonist of Toll-like receptor 7, exhibits antitumor activity against various skin malignancies. We previously reported that IMQ promptly reduced the level of the antiapoptotic Mcl-1 protein and that Mcl-1 overexpression attenuated IMQ-triggered apoptosis in skin cancer cells. Furthermore, IMQ profoundly disrupted mitochondrial function, promoted mitochondrial fragmentation, induced mitophagy, and caused cell death by generating high levels of ROS. However, whether Mcl-1 protects mitochondria from IMQ treatment is still unknown. In this study, we demonstrated that Mcl-1 overexpression induced resistance to IMQ-induced apoptosis and reduced both IMQ-induced ROS generation and oxidative stress in cancer cells. Mcl-1 overexpression maintained mitochondrial function and integrity and prevented mitophagy in IMQ-treated cancer cells. Furthermore, IL-6 protected against IMQ-induced apoptosis by increasing Mcl-1 expression and attenuating IMQ-induced mitochondrial fragmentation. Mcl-1 overexpression ameliorates IMQ-induced ROS generation and mitochondrial fragmentation, thereby increasing mitochondrial stability and ultimately attenuating IMQ-induced cell death. Investigating the roles of Mcl-1 in mitochondria is a potential strategy for cancer therapy development.

## 1. Introduction

Mitochondria are organelles that control the critical mechanisms required for survival, such as bioenergetic processes, intermediate metabolism, reactive oxygen species (ROS) production, calcium buffering, apoptosis regulation, developmental processes, and aging [1,2]. Due to their continuous fission and fusion cycle, mitochondria constantly undergo changes and exhibit a high degree of dynamics to maintain their functions and undergo quality control [3]. An imbalance in fission or fusion results in mitochondrial dysfunction and causes cell death or disease. Evidence indicates that tumorigenesis is mediated by mitochondrial energetics, mitochondrial genome mutations, excessive mitochondrial ROS production, irregular mitochondrial fusion and fission, and dysfunctional mitochondrial mass change [4]. Excessive ROS levels lead to increased cellular and mitochondrial genome mutation frequencies and tumorigenesis rates. However, promoting mitochondrial ROS production can also promote cancer chemotherapy [5,6]. This evidence indicates that the modulation of mitochondria determines the destiny of cancer cells.

Myeloid cell leukemia-1 (Mcl-1) belongs to the Bcl-2 family of proteins and is characterized by its antiapoptotic function. The half-life of Mcl-1 is low, and the expression of Mcl-1 can be stimulated by survival and differentiation signals through the MAPK, PI3K, and JAK signaling pathways [7]. When induced by survival signaling, Mcl-1 promotes survival by inhibiting early-stage caspase activation, which typically leads to cytochrome c release from mitochondria. Interestingly, alternative splicing of Mcl-1 L, which generates the Mcl-1S isoform, leads to apoptosis [8,9]. Moreover, Mcl-1 isoforms can modulate mitochondrial functions, integrity, and dynamics [10,11]. The localization of Mcl-1 in different subcompartments of mitochondria leads to its distinct function. Full-length Mcl-1 located on the outer membrane shows antiapoptotic activity and maintains mitochondrial integrity. The amino-terminus-truncated form of Mcl-1 is located in the matrix and shows no antiapoptotic activity but maintains the inner membrane structure, promotes fusion, and facilitates the assembly of F_0_-F_1_ ATP synthase to promote mitochondrial homeostasis [11]. Many studies have indicated that the expression of Mcl-1 is important for cancer cell survival during chemotherapy [7,12,13,14,15,16,17]. During chemotherapy, a population of chemoresistant cells is naturally selected, and some of these cells overexpress Mcl-1 [17]. Hence, targeting and studying the role of Mcl-1 in cancer cells may aid in the development of cancer therapies.

Imiquimod (IMQ) is a Toll-like receptor 7 (TLR7) agonist used for the in situ treatment of cutaneous squamous cell carcinoma, superficial basal cell carcinoma (BCC), cutaneous malignant melanoma metastases, and precursor lesions of actinic keratosis [18,19]. IMQ is a Th1 immune response mediator and activates TLR7-mediated signaling cascades [20]. IMQ also induces a TLR7-independent response by activating the NLRP3/inflammasome pathway via increased ROS production [21]. Our previous studies indicated that IMQ induces cancer cell death by increasing ROS production, decreasing the mitochondrial membrane potential, and ultimately causing mitochondrion-mediated apoptosis and autophagy [14,15,22,23,24,25]. Additionally, IMQ-induced cell death is associated with mitochondrial dysfunction, mitochondrial fission, and mitophagy [26]. The alleviation of IMQ-induced ROS production may reverse mitochondrial fission to induce fusion and thus further attenuate IMQ-induced mitophagy. Furthermore, the expression of Mcl-1 markedly decreases in cancer cells after IMQ treatment. In contrast, overexpressed Mcl-1 in cancer cells protects these cells from IMQ-induced apoptosis [14]. Recent studies have indicated that the expression of Mcl-1 is important for mitochondrial function and structural integrity [27]. Interestingly, IMQ has been reported to directly target mitochondrial complex I, one of the components of the electron transport chain (ETC), and cause mitochondrial dysfunction [21]. This evidence suggests that the expression of Mcl-1 may play a significant role in IMQ-induced ROS generation, mitochondrial dysfunction, mitochondrial structure, mitophagy, and cell death in cancer cells. However, the role of Mcl-1 in IMQ-induced cell death, as well as its protection-conferring mechanism and physiological function in cancer cells, remains unclear.

In this study, we provide evidence that Mcl-1 not only protects IMQ-treated cancer cells from apoptosis but also maintains mitochondrial function and integrity to prevent IMQ-induced mitochondrial damage, ROS overproduction, and mitophagy. IL-6 enhances Mcl-1 expression and plays a protective role in preventing IMQ-induced apoptosis and decreasing IMQ-induced mitochondrial damage. These results help us understand the protective role of the IL-6/Mcl-1 axis in IMQ-induced death and suggest that targeting the IL-6/Mcl-1 axis may promote the antitumor activity of IMQ.

## 2. Materials and Methods

### 2.1. Reagents and Antibodies 

IMQ was purchased from InvivoGen (San Diego, CA, USA). G418, trypan blue, oligomycin, FCCP, antimycin A1, and rotenone were purchased from Sigma (St. Louis, MO, USA). Antibodies against Mcl-1(#SC-819) were purchased from Santa Cruz Biotechnology (Santa Cruz, CA, USA). Antibodies against Mfn-1(#14739), PINK1(#6946), DRP1(#8570T), phosphorylated S616-DRP1 (#4494T), and TOM20(#42406) were purchased from Cell Signaling Technology (Danvers, MA, USA). The antibody against β-actin (#MA1-140) was purchased from Thermo Fisher Scientific (Grand Island, NY, USA). Recombinant human IL-6 protein was purchased from PeproTech (Cranbury, MJ, USA). Lipofectamine 2000, MitoTracker Red CMXRos, MitoTracker Green FM, LysoTracker Red DND-99, propidium iodide (PI), 5-(and-6)-chloromethyl-2′,7′-dichlorodihydrofluorescein diacetate, acetyl ester (CM-H_2_DCFDA), dihydroethidium (DHE), and MitoSOX Red were purchased from Invitrogen (Carlsbad, CA, USA). 

### 2.2. Cell Culture 

The BCC/KMC-1 (BCC) human basal cell carcinoma cell line was kindly provided by Dr. Yu (Department of Dermatology and Graduate Institute of Clinical Medicine, Faculty of Medicine, Kaohsiung Medical University, Taiwan) and established as previously described [28]. The AGS human gastric adenocarcinoma cell line was obtained from the ATCC (CRL-1739, Manassas, VA, USA) and cultured in RMPI and F-12K media (Gibco, Carlsbad, CA, USA). To generate BCC and AGS clones overexpressing Mcl-1, a DNA fragment containing the full-length human Mcl-1 open reading frame was inserted into the pcDNA3.1 mammalian expression vector (Invitrogen). This pcDNA3.1-Mcl-1 plasmid was then transfected into BCC and AGS cells via Lipofectamine 2000 (Invitrogen). After transfection, the cells were treated with G418 (800 µg/mL, Sigma) for 48 h to select stably transfected clones. 10H and 12A cells showed BCC-based Mcl-1-overexpressing clones. AGS IV was AGS-based Mcl-1-overexpressing clones. The level of Mcl-1 overexpression in these clones was confirmed through immunoblotting with anti-human Mcl-1 antibodies. Mcl-1-overexpressing BCC [14] and AGS cells were cultured in medium containing 800 µg/mL G418. All media were supplemented with 10% fetal bovine serum (FBS), and all the cells were maintained at 37 °C in a humidified 5% CO_2_ incubator.

### 2.3. Cellular, Mitochondrial ROS and Lipid ROS Assays

Cancer cells were seeded in 6-well plates (2 × 10^5^ cells/well) and allowed to reach 70–80% confluency. The cells were incubated with 1 mL of medium containing 50 µg/mL IMQ for 4 h. The cells were stained with 1 μM CM-H2DCFDA for total ROS detection and with 2.5 μM DHE and MitoSOX Red for cellular and mitochondrial superoxide detection for 30 min. The stained cells were analyzed with a BD FACSCaliburTM cytometer (Beckman Coulter, Brea, CA, USA). The cells were treated with 50 µg/mL IMQ for 24 h. The lipid peroxidation level was assessed with a lipid peroxidation (MDA) colorimetric/fluorometric assay kit (BioVision, Milpitas, CA, USA) following the manufacturer’s protocol.

### 2.4. Cell Viability and DNA Content Assays

Cancer cells were seeded in 6-well plates (2 × 10^5^ cells/well) and allowed to reach 70–80% confluency. The cells were treated with 1 mL of medium supplemented with 50 µg/mL IMQ for 24 h. Cell viability was assessed with a trypan blue staining assay following the manufacturer’s protocol (Sigma). Apoptotic cells were quantified by assessing the population of subG1 stained with PI and analyzed via BD FACSCaliburTM cytometer (Beckman Coulter).

### 2.5. Immunoblotting

Cancer cells were seeded in 6-well plates (2 × 10^5^ cells/well) and allowed to reach 70–80% confluency. The cells were treated with 1 mL of medium supplemented with 50 µg/mL IMQ for 12 or 24 h. The cells were harvested in PRO-PREP protein extraction solution (iNtRON, Taipei, Taiwan) containing a protease inhibitor cocktail and vigorously shaken at 4 °C for 30 min, followed by centrifugation. The supernatants were collected, and the protein concentrations were subsequently determined by using Bio-Rad BCA reagent (Bio-Rad, Hercules, CA, USA). A total of 30 μg of each sample lysate was subjected to electrophoresis on SDS–polyacrylamide gels, then electroblotted onto PVDF membranes. After being blocked with 5% BSA in TBST, the membranes were incubated with primary antibodies in TBST at 4 °C overnight. The membranes were washed 4 times and incubated with horseradish peroxidase (HRP)-conjugated goat anti-mouse or rabbit IgG (Upstate, Billerica, MA, USA) for 2 h. After washing with TBST 4 times, the blots were incubated for 1 min with SuperSignal West Pico ECL reagent (Pierce Biotechnology, Rockford, IL, USA), and chemiluminescence was detected via exposure to Kodak-X-Omat film (Eastman Kodak Company, Rochester, NY, USA).

### 2.6. Mitochondrial Oxygen Consumption Assay

Cancer cells were seeded in 6-well plates (2 × 10^5^ cells/well) and allowed to reach 70–80% confluency. After suspension, the cells were incubated with 50 µg/mL IMQ, then immediately transferred to an OROBOROS Oxygraph-2k (O2k) instrument (Oroboros Instruments GmbH, Innsbruck, Tyrol, Austria) for respiration measurement. They were sequentially treated with 2 µM oligomycin, followed by titration with 0.5 µM FCCP, then 2 µM antimycin A1, and rotenone, to assess the oxygen consumption rate (OCR) in both control and IMQ-treated cells.

### 2.7. Confocal Imaging for Colocalization of Mcl-1 and Mitochondria 

Cancer cells were seeded in 6-well plates containing coverslips (2 × 10^5^ cells/well) and allowed to reach 70–80% confluency. After treatment with 50 µg/mL IMQ in 1 mL of medium for 24 h, the cells were stained with 100 nM MitoTracker Red CMXRos for 30 min, then fixed with 1% formaldehyde in PBS at 4 °C overnight. The fixed cells were blocked with 2% bovine serum albumin (BSA, Gibco), incubated with antibodies against Mcl-1 in PBS containing 0.2% Triton X-100 (PBST) at 4 °C overnight, and incubated with a goat Alexa Fluor 488 dye-conjugated anti-rabbit IgG antibody (Thermo, Carlsbad, CA, USA). After being washed with PBST, the cells were mounted using an antifade, 4′,6-diamidino-2-phenylindole (DAPI)-containing, water-based mounting medium (Vector Laboratories, Burlingame, CA, USA). The fluorescence images were acquired with a confocal microscope (Olympus, FV1000D, Tokyo, Japan).

### 2.8. Confocal Imaging for Assessment of Mitochondrial Morphology

Cancer cells were seeded in 6-well plates containing coverslips (2 × 10^5^ cells/well) and allowed to reach 70–80% confluency. After treatment with 50 µg/mL IMQ in 1 mL of medium, the cells were cultured for 24 h. To assess mitochondrial morphology, the cells were stained with 100 nM MitoTracker Red CMXRos or 100 nM MitoTracker Green FM for 30 min. Fluorescence images were acquired via a confocal microscope (Olympus, FV1000D). Mitochondria were categorized on the basis of their morphology via CellSens Software (Olympus). The mitochondria that were wholly fragmented or permeabilized were classified as fragmented mitochondria (Class 1), those with clear networks were labeled as tubular mitochondria (Class 3), and those with other shapes were categorized as intermediate mitochondria (Class 2). The proportions of mitochondria with these specific morphologies were calculated as percentages of the total number of examined cells (100 cells per experiment).

### 2.9. Confocal Imaging for the Colocalization of Mitochondria and Lysosomes

Cancer cells were seeded in 6-well plates containing coverslips (2 × 10^5^ cells/well) and allowed to reach 70–80% confluency. The cells were preincubated with 100 nM MitoTracker Green FM for 30 min, followed by 50 µg/mL IMQ in 1 mL of medium for 24 h, then stained with 1 µM LysoTracker Red DND-99 for 30 min. The colocalization rates of mitochondria with lysosomes were monitored via confocal microscopy and analyzed with Olympus FV10-ASW V4.2 software (Olympus).

### 2.10. Statistical Analysis

Three independent experiments were repeated for all the assays, and all the assays were performed in duplicate or triplicate. The data were analyzed using one- or two-way ANOVA, and differences were considered significant when ** p* < 0.05, *** p* < 0.01, and **** p* < 0.001.

## 3. Results

### 3.1. Mcl-1 Overexpression Diminished IMQ-Induced Cancer Cell Death

We previously reported that overexpressed Mcl-1 in BCC 10H and 12A clones caused IMQ resistance [14]. To confirm that the IMQ resistance effect of Mcl-1 overexpression was not cell-specific, we generated another Mcl-1-overexpressing clone from the AGS gastric cancer cell line and named it AGS IV. Compared with control BCC and AGS cells, stable Mcl-1-overexpressing clones 10H, 12A, and AGS IV presented consistently higher Mcl-1 protein levels (Figure 1A). Through trypan blue staining assays, we consistently observed greater viability of the 10H, 12A, and AGS IV cells than of control counterparts after IMQ treatment (Figure 1B and Appendix A). We subsequently investigated the impact of Mcl-1 on IMQ-induced apoptosis by conducting a DNA content assay, which allowed us to assess the rates of apoptotic responses. Compared with control cells, 10H, 12A, and AGS IV cells presented much lower sub-G1 DNA abundance after IMQ treatment (Figure 1C and Appendix A). These results indicate that Mcl-1 overexpression confers resistance to IMQ-induced apoptosis in various cancer cells.

### 3.2. Overexpressed Mcl-1 Moderated IMQ-Induced Oxidative Stress Levels in Cancer Cells

Previously, we demonstrated that ROS play an important role in IMQ-induced apoptosis [23,25]. Next, we examined whether Mcl-1-overexpressing BCC and AGS cells decrease IMQ-induced ROS generation. The results showed that ROS production was markedly increased in both types of control cells. In contrast, ROS production in the 10H, 12A, and AGS IV cells was significantly lower than that in their control counterparts (Figure 2A and Appendix A). Similarly, the levels of cellular and mitochondrial superoxide produced after IMQ treatment were significantly greater in the control cells than in the Mcl-1-overexpressing cells (Figure 2B,C and Appendix A). In addition, the level of lipid peroxide, a marker of oxidative stress, was lower in the Mcl-1-overexpressing cells than in the control cells (Figure 2D). As shown in Appendix A, the populations of endogenous and overexpressed Mcl-1 proteins overlapped, as shown by MitoTracker Red CMXRos staining of organelles, indicating that the majority of overexpressed Mcl-1 proteins were located in mitochondria. Therefore, mitochondrial Mcl-1 may not only attenuate IMQ-induced ROS production but also reduce cellular and mitochondrial superoxide production to alleviate oxidative stress in BCC and AGS cells.

### 3.3. Mcl-1 Overexpression Increased the Mitochondrial Oxygen Consumption Rate in Cancer Cells

Mitochondria are significant sources of ROS within mammalian cells. We previously reported that IMQ disrupted mitochondrial function and structure [26]. Mitochondrial respiratory quality represents mitochondrial functionality. We used an OROBOROS O2k instrument to investigate mitochondrial functions by monitoring the oxygen consumption rate (OCR) in control and Mcl-1-overexpressing cells with or without IMQ treatment. As shown in Figure 3 and Appendix A, IMQ decreased the basal respiration (routine, R), leakage (L), and maximal oxygen consumption (electron transfer system, ETS or E) of mitochondria in cancer cells. No differences in the residual oxygen consumption (ROX state) were observed between the control and IMQ-treated cells. The parameter of routine subtract leakage (R-L) indicates the ATP-linked respiratory rate. We previously presented evidence indicating that IMQ induced metabolic stress and caused ATP depletion [15]. After IMQ treatment, the ATP-linked respiratory rate was markedly decreased in IMQ-treated cells. These findings suggest that the ATP depletion caused by IMQ treatment corresponds to retarded ATP-linked respiration. The parameter of the ETS subtraction routine (E-R), called reverse capacity, indicates the ability of mitochondria to withstand physiological stress [29]. In the IMQ-treated cells, the E-R parameter was lower than that in the control cells. Interestingly, Mcl-1-overexpressing BCC 12A (Figure 3), 10H (Appendix A), and AGS IV cells (Appendix A) had a higher mitochondrial respiratory rate than the control BCC and AGS cells. Importantly, although mitochondrial function decreased in both cancer cell lines after IMQ treatment, mitochondrial function was still maintained at a relatively high level in Mcl-1-overexpressing cells. Taken together, these findings indicate that Mcl-1 overexpression may increase the mitochondrial OCR and increase mitochondrial capacity in cancer cells in order to endure physiological stresses.

### 3.4. Mcl-1 Overexpression Stabilized Mitochondrial Dynamics in Cancer Cells after IMQ Treatment

When mitochondria are damaged from stressors such as ROS, the delicate balance in mitochondrial dynamics is disrupted, leading to an increase in mitochondrial fission [3]. To evaluate mitochondrial morphology, MitoTracker Red CMXRos was used to observe mitochondria. We also classified the lengths of mitochondria in BCC, AGS control, and Mcl-1-overexpressing cells. The class 1 mitochondria were short and fragmented, the class 2 mitochondria were intermediate mitochondria, and the class 3 mitochondria were long and tubular. As shown in Appendix A, Mcl-1-overexpressing 12A and AGS IV cells contained more long, tubular mitochondria than BCC and AGS control cells. Under the stress conditions induced by IMQ, the number of short and fragmented mitochondria significantly increased, and the number of tubular mitochondria decreased in the control cells. However, Mcl-1-overexpressing cancer cells contained more abundant tubular mitochondria than fragmented mitochondria (Figure 4A and Appendix A). Mcl-1-overexpressing 12A and AGS IV cells maintained Mcl-1 protein stability after IMQ treatment (Figure 4B). The regulation of mitochondrial fission and fusion is orchestrated by specific mitochondrial dynamin-related GTPases, including Mfn1/2, OPA1, and DRP1 [30]. As shown in Figure 4B, IMQ-induced expression of p-DRP1(Ser616), a mitochondrial fission marker, in BCC and AGS control cells was greater than that in Mcl-1-overexpressing cancer cells. In contrast, IMQ decreased the expression of mitochondrial fusion marker Mfn1 in BCC and AGS control cells but not in Mcl-1-overexpressing cancer cells. These results indicate that Mcl-1 overexpression not only stabilizes mitochondrial dynamics but also elevates the rate of mitochondrial fusion in IMQ-treated cancer cells.

### 3.5. Mcl-1 Overexpression Inhibited IMQ-Induced Mitophagy in Cancer Cells

Mitophagy is a form of selective autophagy that specifically targets and removes damaged or dysfunctional mitochondria [31]. After IMQ stimulation, the number of lysosomes increased, and they colocalized with fragmented mitochondria in BCC and AGS cells. In contrast, the number of colocalized mitochondria and lysosomes was significantly decreased in 10H, 12A, and AGS IV cells (Figure 5A and Appendix A). These results indicate that Mcl-1 overexpression might reduce IMQ-induced mitophagy. To confirm these results, the expression of TOM20 and PINK1 in IMQ-treated Mcl-1-overexpressing and control cells was measured. TOM20 degradation is a marker of mitophagy [32], and measures of TOM20 expression are used to quantify mitochondrial mass [33]. PINK1 plays a specific role in the ubiquitin targeting of damaged mitochondria by facilitating the Parkin-mediated ubiquitination of substrates on mitochondria. Therefore, a high PINK1 level is a marker of mitophagy [33]. We previously showed that IMQ not only disrupted the balance of mitochondrial dynamics but also triggered mitophagy in skin cancer cells [26]. In both BCC and AGS control cells, IMQ treatment led to a decrease in the expression of TOM20, as illustrated in Figure 5B. Surprisingly, we found that TOM20 was overexpressed in 12A and AGS IV cells and that the expression of TOM20 was even increased in both types of IMQ-treated Mcl-1-overexpressing cells. Notably, the expression of mitophagy-related protein PINK1 did not significantly increase after IMQ treatment in 12A and AGS IV cells as compared with untreated 12A and AGS IV cells. Collectively, these findings suggest that IMQ does not trigger mitophagy in cells overexpressing Mcl-1.

### 3.6. Exogenous IL-6 Treatment Increased Mcl-1 Levels and Reduced Cell Death and ROS Generation in IMQ-Treated BCC Cells

Mcl-1 is classified as an antiapoptotic protein that is overexpressed in many types of IL-6-treated cancer cells, including prostate cancer, myeloma, gastric cancer, and cholangiocarcinoma cells, but not in control cells [34]. Importantly, Mcl-1 protein expression can be upregulated by IL-6 or exogenous IL-6 treatment through PI3K/Akt activation to prevent UV and photodynamic therapy-induced apoptosis in BCC cells [35,36]. Therefore, we hypothesized that exogenous IL-6 supplementation may increase the expression of Mcl-1, protecting BCC cells against IMQ-induced apoptosis. We found that pretreatment with 10 ng/mL human recombinant IL-6 induced high Mcl-1 protein expression in BCC cells (Figure 6A). Interestingly, higher Mcl-1 protein levels were observed in IMQ-treated BCC cells after IL-6 pretreatment than in BCC cells treated with only IMQ. Additionally, IL-6 pretreatment not only protected BCC cells from IMQ-induced cell death (Figure 6B) but also reduced the ROS generation rate in IMQ-treated BCC cells (Figure 6C). We investigated whether IL-6-induced Mcl-1 overexpression attenuates IMQ-induced mitochondrial fragmentation. As shown in Figure 6D, compared with that in IMQ-treated cells, the rate of mitochondrial fragmentation was lower in IL-6-pretreated cells. Therefore, IL-6 may upregulate Mcl-1 protein expression to attenuate IMQ-triggered ROS production and protect BCC cells from IMQ-induced cell death.

## 4. Discussion

Mcl-1, a prosurvival protein, is a crucial member of the Bcl-2 family and has been shown to be overexpressed in various cancer types, including ovarian, cervical, hepatocellular, pancreatic, non-small cell lung, and testicular germ cell cancers and melanomas [37]. Our previous study indicated that Mcl-1 may play a vital role in protecting skin cancer cells against IMQ-induced apoptosis [14]. In the present study, we demonstrated that Mcl-1 overexpression protected not only skin cancer cells but also gastric cancer cells from IMQ-induced cytotoxicity (Figure 1). These data further support the hypothesis that Mcl-1 overexpression in cancer cells contributes to IMQ resistance.

IMQ-induced ROS production has been measured in various types of cells [22,23,26,38]. Mitochondrial complex I and cytosolic protein NQO2 are the major targets of IMQ and trigger ROS production [21]. ROS are inevitable byproducts of the mitochondrial respiratory chain, and the accumulation of ROS leads to mitochondrial damage; dysfunction; bioenergetic failure; and, ultimately, cell death [39,40]. We previously showed that Mcl-1 may play an important role in protecting cancer cells from IMQ-induced cell death [14]. Nuclear factor erythroid 2-related factor 2 (Nrf2) not only plays a definitive role in the regulation of oxidative stress but also contributes to chemoresistance, which Mcl-1 also promotes in multiple types of cancer [41,42,43,44,45]. In this study, we found that IMQ-induced production of ROS and mitochondrial superoxide was significantly abolished in Mcl-1-overexpressing BCC and AGS cancer cells (Figure 2 and Appendix A). Nrf2 directly inhibits apoptosis by upregulating the expression of BCL-2 and BCL-XL [46]. However, the upregulation of BCL-2 also enhances Nrf2 activity [47]. It is possible that Mcl-1 may increase Nrf2 activation to protect cancer cells from IMQ-induced oxidative stress. The crosstalk between Mcl-1 and Nrf2 in IMQ-induced cell death and oxidative stress is a significant concern and is unknown. We will further investigate these issues. Compared with WT mitochondria, mitochondria lacking Mcl-1 present lower mitochondrial respiratory chain efficiency and increased ROS production [11,48]. As shown in Appendix A, we found that the colocalization rate of Mcl-1 with mitochondria was significantly greater in 12A and AGS IV cells than in BCC and AGS control cells. The location of Mcl-1 in the mitochondria determines mitochondrial dynamics, mitochondrial functions, and cell survival [11]. Therefore, Mcl-1 may stabilize mitochondrial homeostasis to against IMQ treatment in cancer cells.

Retarding ETC function leads to the unmanageable production of excessive ROS, subsequently causing mitochondrial dysfunction, including mitochondrial membrane potential collapse and mitochondrial respiratory chain decline [49]. As shown in Figure 3, IMQ inhibited complex I function, blocking the ETC and decreasing mitochondrial respiratory chain function in both control and Mcl-1-overexpressing cells. Notably, the mitochondrial respiratory rate was greater in Mcl-1-overexpressing cells than in the control cells. In parallel with the functional consequences of Mcl-1 overexpression for mitochondrial bioenergetics, including ATP production, cellular respiration, and metabolism, Mcl-1 contributes to normal mitochondrial fission and fusion and modulates the ultrastructure of mitochondrial cristae [50]. Therefore, more stable mitochondrial function contributes to reduced electron leakage and ROS production. These data are consistent with Mcl-1 overexpression-induced moderation of IMQ-induced oxidative stress in cancer cells (Figure 2 and Appendix A). Taken together, these results indicate that Mcl-1 overexpression not only further optimized mitochondrial function but also stabilized the mitochondrial respiratory chain in IMQ-treated cancer cells.

Under steady-state conditions, the fusion and fission frequencies of mitochondria are balanced to preserve the overall morphology and health of the mitochondrial population [3]. Disruption of this balance leads to significant changes in mitochondrial morphology. In our previous study, IMQ disrupted the balance of mitochondrial dynamics, and mitophagy induction was associated with ROS production in skin cancer cells [26]. In this study, IMQ not only triggered mitochondrial fragmentation but also activated DRP1 and decreased the expression of Mfn1, thereby promoting mitochondrial fission. Interestingly, Mcl-1 overexpression reversed IMQ-induced mitochondrial fragmentation and the expression of Mfn1 (Figure 4), and it was accompanied by reduced ROS production, further stabilizing mitochondrial function (Figure 2 and Figure 3). Collectively, our results indicate that overexpressed Mcl-1 might be localized not only to the outer mitochondrial membrane but also to the mitochondrial matrix, where it preserves the normal structure of the inner mitochondrial membrane, facilitates mitochondrial fusion, and reverses IMQ-triggered mitochondrial fragmentation and dysfunction to attenuate cell death in cancer cells.

Mitophagy is an oxidative stress-mediated mechanism that preserves the integrity and quality of the mitochondrial population [31]. Polarized and damaged mitochondria no longer sustain the mitochondrial fission and fusion cycles and tend to undergo mitochondrial fission [26]. Short and fragmented mitochondria can be easily engulfed by the isolation membrane and undergo autophagy. In contrast, when ROS-induced mitochondrial fragmentation is ameliorated, mildly damaged mitochondria can fuse with healthy mitochondria, attenuating injury. Mcl-1 has been reported to hinder mitophagy by preventing the translocation of Parkin to mitochondria with depolarized membranes [51]. Similarly, the expression of TOM20 was much greater in 12A and AGS IV cells than in control cells and was unaffected by IMQ treatment, indicating that IMQ induced low rates of mitophagy in Mcl-1-overexpressing cells (Figure 5 and Appendix A). Therefore, overexpressed Mcl-1 inhibits IMQ-induced mitophagy, and this effect is accompanied by minor mitochondrial dysfunction.

IL-6 contributes to the differentiation and proliferation of different types of malignant cells [52]. Notably, IL-6 upregulates Mcl-1 expression through the JAK/STAT3 signaling pathway, which not only induces the proliferation and survival of multiple myeloma cells but also increases mitochondrial bioenergy production and confers protection against mitochondrial dysfunction in cortical neurons [53,54]. In addition, the overexpression of IL-6 induced Mcl-1 overexpression and inhibited apoptosis in basal cell carcinoma [36]. Therefore, IL-6 may play a critical role in the regulation of Mcl-1. In this study, we demonstrated that IL-6-induced Mcl-1 overexpression not only alleviated IMQ-induced ROS generation and mitochondrial fragmentation but also attenuated IMQ-triggered death in BCC cells (Figure 6). These results suggest that the protective effects of IL-6 are associated with IL-6-induced Mcl-1 overexpression and mitochondrial function.

## 5. Conclusions

In this study, we found that Mcl-1 mitigated IMQ-induced oxidative stress and consequently protected cancer cells from death via IMQ. Furthermore, the decreased stability of Mcl-1 induced by IMQ is associated not only with the production of ROS but also with the maintenance of mitochondrial integrity in cancer cells. Mcl-1 overexpression enhanced mitochondrial fusion and stabilized the mitochondrial structure, providing cytoprotective effects in IMQ-treated cancer cells. Investigating the roles of Mcl-1 in mitochondria may lead to its designation as a potential target for IMQ-related cancer therapy.

## Figures and Tables

**Figure 1 cancers-16-03060-f001:**
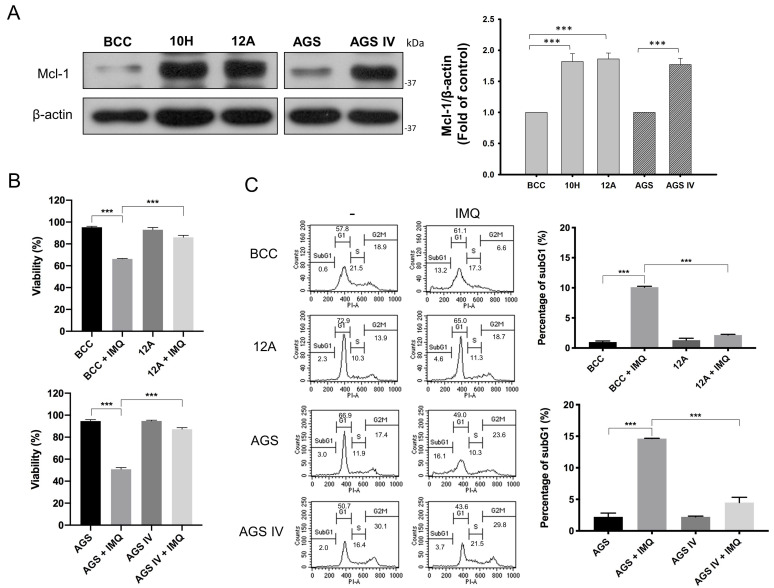
Mcl-1 overexpression in BCC and AGS cells reduced IMQ-induced cell death and apoptosis. (**A**) Mcl-1 was overexpressed in 10H, 12A and AGS IV cells. The expression levels of Mcl-1 and β-actin were measured by immunoblotting. The densitometric analysis results are presented as the ratio of the Mcl-1 protein level to the β-actin protein level. Mcl-1 overexpression attenuated IMQ-induced cell death (**B**) and apoptosis (**C**) in 12A and AGS IV cells. 12A and AGS IV cells were treated with 50 µg/mL IMQ for 24 h. Cell viability was analyzed with a trypan blue staining assay (**B**). Cell cycle analysis was performed with PI staining, then analyzed by flow cytometry (**C**). The data are expressed as the mean ± S.E.M. of three independent experiments. The statistical results were analyzed via two-way ANOVA. *** *p* < 0.001.

**Figure 2 cancers-16-03060-f002:**
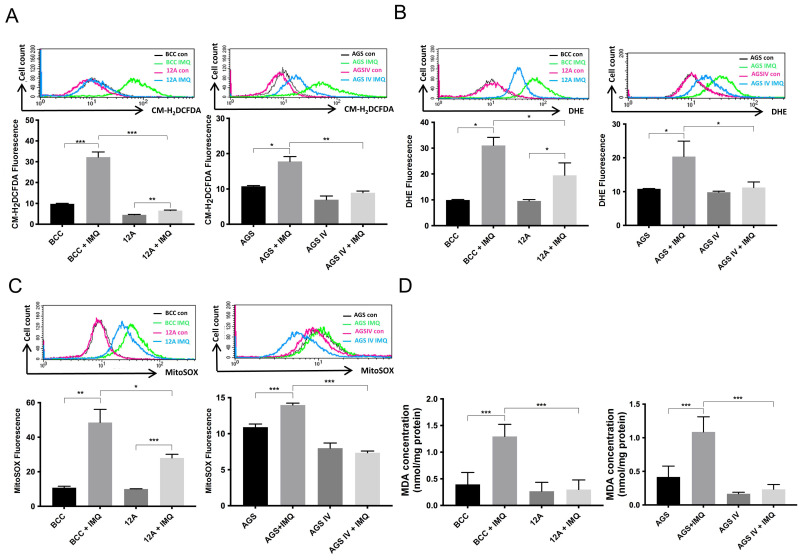
Mcl-1 overexpression reduced IMQ-induced increases in total ROS, cellular and mitochondrial superoxide, and lipid peroxide levels in cancer cells. The cells were treated with 50 µg/mL IMQ for 4 h. After IMQ treatment, the cells were stained with (**A**) CM-H2DCFDA (1 µM) for total ROS measurements. (**B**) DHE (2.5 µM) was used for cellular superoxide measurements, and (**C**) MitoSOX (2.5 µM) red was used for mitochondrial superoxide measurements, which were obtained 30 min after staining via flow cytometry. The cells were treated with 50 µg/mL IMQ for 24 h and harvested, after which the lipid peroxide level was measured with a lipid peroxidation (MDA) colorimetric assay kit (**D**). The data are expressed as the mean ± S.E.M. of three independent experiments. The results were statistically analyzed by two-way ANOVA. * *p* < 0.05, ** *p* < 0.01, and *** *p* < 0.001.

**Figure 3 cancers-16-03060-f003:**
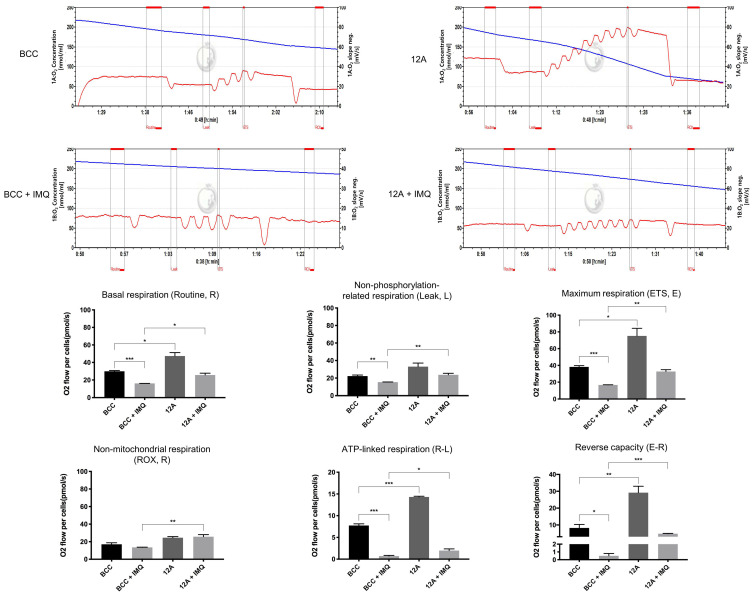
Mcl-1 overexpression increased the mitochondrial oxygen consumption rate but partially counteracted the IMQ-induced decrease in the mitochondrial oxygen consumption rate in BCC cells. BCC control and 12A cells were sequentially treated with 50 µg/mL IMQ, 2 µM oligomycin, 0.5 µM FCCP (titration concentration), 2 µM antimycin A1, and rotenone. The different mitochondrial OCRs were subsequently evaluated with an O2k respirometer. The data are expressed as the mean ± S.E.M. of three independent experiments. The results were statistically analyzed by two-way ANOVA. * *p* < 0.05, ** *p* < 0.01, and *** *p* < 0.001.

**Figure 4 cancers-16-03060-f004:**
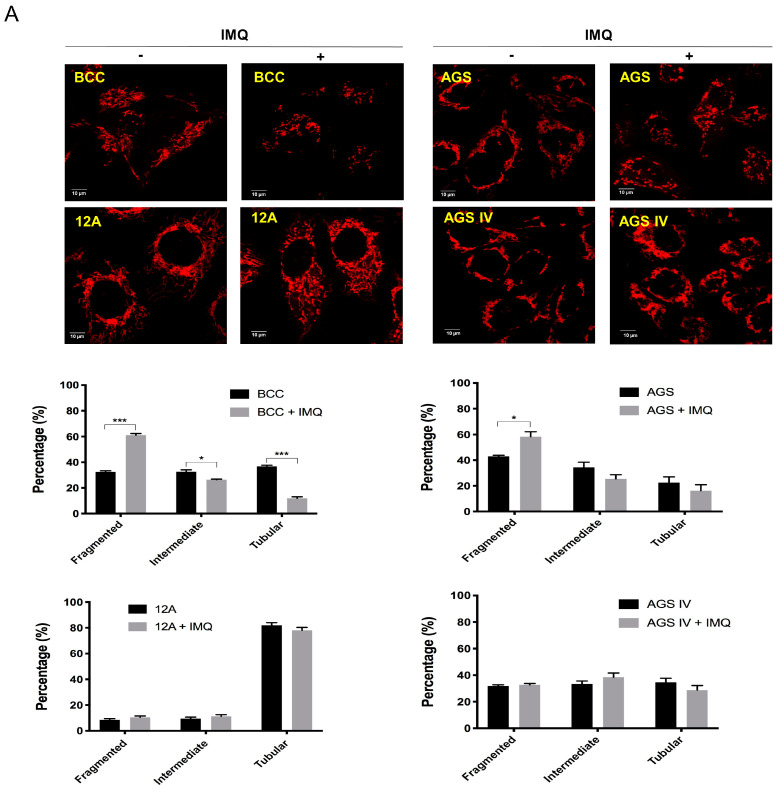
Mcl-1 overexpression led to reduced IMQ-induced imbalance in mitochondrial dynamics in cancer cells. (**A**) Mcl-1 overexpression increased the mitochondrial length in IMQ-treated cancer cells. (**B**) Mcl-1 overexpression regulated the expression of fission- and fusion-related markers in IMQ-treated cancer cells. (**A**) After 50 µg/mL IMQ treatment for 24 h, cancer cells were stained with 100 nM MitoTracker Red CMXRos for 30 min. Mitochondrial morphology was observed by confocal microscopy, and the number of mitochondria in each morphological class was determined via CellSens Software (Olympus). Scale bars, 10 μm. (**B**) After 50 µg/mL IMQ treatment for 12 h, cancer cells were harvested, and the expression of fission and fusion markers was measured via immunoblotting. The results were statistically analyzed by two-way ANOVA. * *p* < 0.05, ** *p* < 0.01, and *** *p*< 0.001.

**Figure 5 cancers-16-03060-f005:**
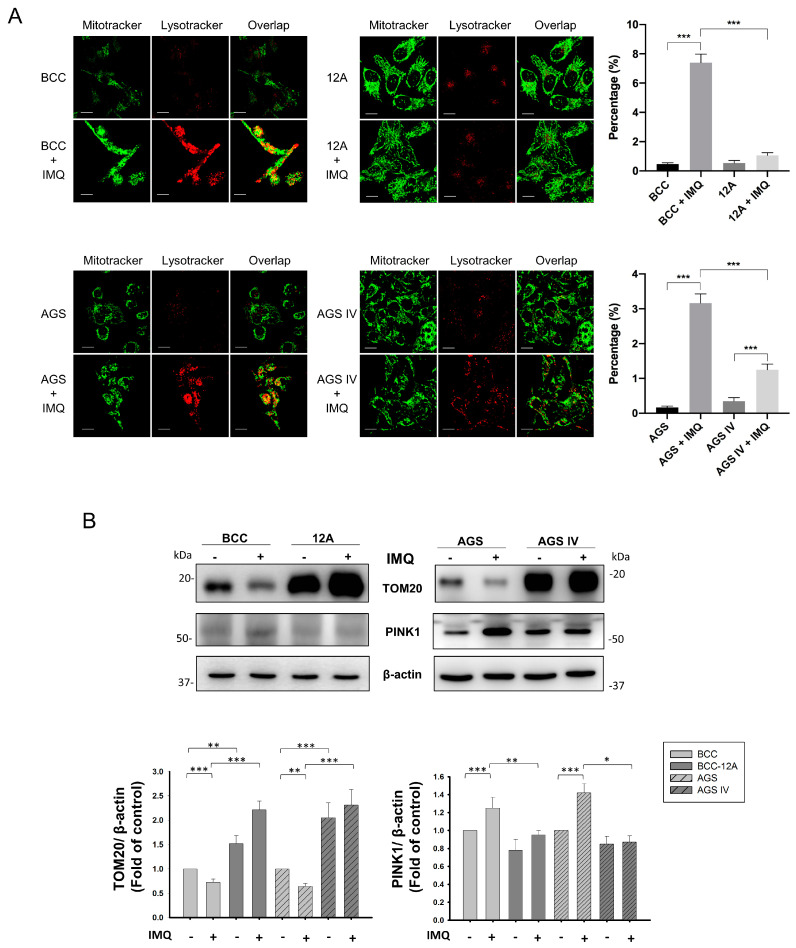
Mcl-1 overexpression modulated IMQ-induced mitophagy in cancer cells. Mcl-1 overexpression not only decreased IMQ-induced colocalization of lysosomes and mitochondrial fragmentation (**A**) but also decreased IMQ-induced PINK1 expression in cancer cells (**B**). (**A**) Cells were preincubated with 100 nM MitoTracker Green FM for 30 min, followed by 50 µg/mL IMQ treatment for 24 h and staining with 1 µM LysoTracker Red DND-99 for 30 min. The colocalization rates of mitochondria with lysosomes were monitored via confocal microscopy and analyzed with Olympus FV10-ASW V4.2 software (Olympus). Scale bars, 10 μm. (**B**) In preparation for immunoblotting, the cells were treated with 50 µg/mL IMQ for 12 h, and the levels of TOM20, PINK1, and β-actin were measured by immunoblotting. The results were statistically analyzed by two-way ANOVA. * *p* < 0.05, ** *p* < 0.01, and *** *p* < 0.001.

**Figure 6 cancers-16-03060-f006:**
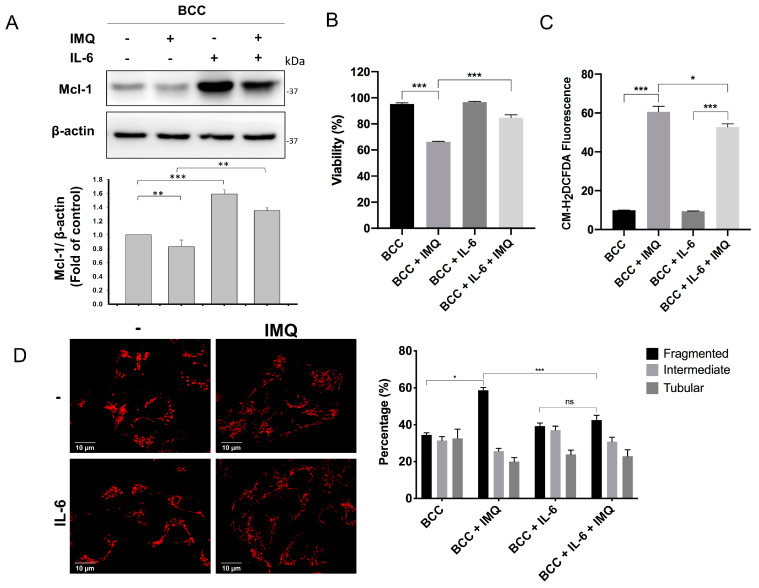
IL-6 increased Mcl-1 overexpression and decreased IMQ-induced death in BCC cells. (**A**) IL-6 treatment upregulated the expression of Mcl-1 in BCC cells. (**B**,**C**) IL-6 treatment decreased IMQ-induced cell death and ROS production rates in BCC cells. (**D**) IL-6 treatment decreased IMQ-induced mitochondrial fragmentation. BCC cells were pretreated with 10 ng/mL IL-6 for 4 h and 50 µg/mL IMQ for 24 h. (**A**) The expression levels of Mcl-1 and β-actin were measured by immunoblotting. (**B**) Cell viability was analyzed with a trypan blue staining assay. (**C**) ROS productions were stained with 1 µM CM-H2DCFDA and analyzed by flow cytometry. (**D**) The mitochondrial morphologies were stained with 100 nM MitoTracker Red CMXRos for 30 min, monitored via confocal microscopy, and further analyzed via CellSens Software (Olympus). Scale bars, 10 μm. The results were statistically analyzed by one-way ANOVA. * *p* < 0.05, ** *p* < 0.01, and *** *p* < 0.001, ns means not significant.

## Data Availability

All data generated or analyzed during this study are included in this published article and Appendix A.

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
