# Peer review of "The Protective Effects of Mcl-1 on Mitochondrial Damage and Oxidative Stress in Imiquimod-Induced Cancer Cell Death"

_cancers, 2024, doi:10.3390/cancers16173060_

Round 1

Reviewer 1 Report

Comments and Suggestions for Authors

Review of the manuscript entitled

„The protective effects of Mcl-1 on the mitochondria damage and oxidative stress in imiquimod-induced cancer cell death”

by Shu-Hao Chang and Kai-Cheng Chuang et al.

Manuscript draft entitled „The protective effects of Mcl-1 on the mitochondria damage and oxidative stress in imiquimod-induced cancer cell death” by Shu-Hao Chang and Kai-Cheng Chuang et al., presents the protective role of Mcl-1 against imiquimod (IMQ)-induced cancer cell death. IMQ is a chemotherapeutic agent used to treat various cancers, including superficial basal cell carcinoma (BCC). The authors studied mainly two cancer cell models including BCC/KMC-1 (BCC cell line) and AGS (gastric adenocarcinoma cell line) comparing cells with basal and overexpressed Mcl-1. The main finding is that Mcl-1 overexpressing cells were less sensitive toward IMQ-induced cell death which was associated with lower oxidative stress, increased mitochondrial oxygen consumption, reduced mitochondrial dynamic imbalance and inhibited mitophagy. Moreover, the authors confirmed that BCC cell line with basal expression of Mcl-1 was protected against IMQ-induced cell death when cells were pretreated with exogenous IL-6 which upregulates Mcl-1 level. The study is well-planned and coherent. In my opinion, the work should be published after major corrections enlisted below.

#1 Please include information in Materials and Methods how many cells were seeded for each experiment. Please include information about the size of the cell culture plates and the volume of the medium.

#2 Please add the information in Materials and Methods about the generation of Mcl-1 overexpressing cells. It appears in short version in Fig. 1 legend.

#3 Please verify the concentrations of CM-H2DCFDA (1 mM) and DHE (2.5 mM) in Materials and Methods.

#4 Please include the ID numbers of antibodies used in the study.

#5 Please add the bar graphs for each Western blot experiment.

#6 The Mcl-1 overexpressing clone H10 was used only in Western blot studies showing increased expression of Mcl-1. The manuscript would benefit from the performance of additional experiments including cell viability, oxidative stress, mitochondrial oxygen consumption, mitochondrial dynamics and mitophagy which would strengthen the results.

#7 Please correct the title in 3.4 section.

#8 Please correct the legends in Fig. 4 and Fig. 5.

#9 The Conclusion section could be improved.

#10 The manuscript requires minor language correction.

Comments on the Quality of English Language

The manuscript requires minor language correction.

Author Response

Response to Reviewer #1’ comments:

Thank you for your good comments and suggestions. According to your opinions, we have revised the manuscript, and the answers are below:

Question 1: Please include information in Materials and Methods how many cells were seeded for each experiment. Please include information about the size of the cell culture plates and the volume of the medium.

Response 1: Thank you for your suggestion. We included the cell number, the size of the cell culture plates, and the volume of the medium in each experiment section 2.3, 2.4, 2.5, 2.6, 2.7, 2.8 and 2.9 of Materials and Methods. (Page 4 and 5)

Question 2: Please add the information in Materials and Methods about the generation of Mcl-1 overexpressing cells. It appears in short version in Fig. 1 legend.

Response 2: Thank you for kindly suggestion. We added the information in Materials and Methods section 2.2 about the generation of Mcl-1 overexpressing cells as below: To generate BCC and AGS clones overexpressing Mcl-1, a DNA fragment containing the full-length human Mcl-1 open reading frame was inserted into the mammalian expression vector pcDNA3.1 (Invitrogen). This pcDNA3.1-Mcl-1 plasmid was then transfected into BCC and AGS cells via Lipofectamine 2000 (Invitrogen). After transfection, the cells were treated with G418 (800 µg/ml, Sigma) for 48 hours to select stably transfected clones. 10H and 12A cells showed BCC-based Mcl-1-overexpressing clones. AGS IV was AGS-based Mcl-1-overexpressing clones. (Page 3, Line 146-152)

Question 3: Please verify the concentrations of CM-H2DCFDA (1 mM) and DHE (2.5 mM) in Materials and Methods.

Response 3: Thank you for your careful suggestions. We corrected the concentrations of CM-H2DCFDA and DHE in section 2.3 as below: The cells were stained with 1 μM CM-H2DCFDA for total ROS detection and with 2.5 μM DHE and MitoSOX Red for cellular and mitochondrial superoxide detection for 30 min. (Page 4, Line 161-163)

Question 4: Please include the ID numbers of antibodies used in the study.

Response 4: Thank you for your kindly suggestions. We included the ID numbers of antibodies in section “2.1. Reagents and antibodies” as described below: Antibodies against Mcl-1(#SC-819) were purchased from Santa Cruz Biotechnology (Santa Cruz, CA, USA). Antibodies against Mfn-1(#14739), PINK1(#6946), DRP1(#8570T), phosphorylated S616-DRP1 (#4494T) and TOM20(#42406) were purchased from Cell Signaling Technology (Danvers, MA, USA). The antibody against β-actin (#MA1-140) was purchased from Thermo Fisher Scientific (Grand Island, NY, USA). (Page 3, Line 129-133)

Question 5: Please add the bar graphs for each Western blot experiment.

Response 5: We added the bar graphs for each Western blot including Figure 1A, 4B, 5B and 6A. (Page 6, 10, 11 and 12)

Question 6: The Mcl-1 overexpressing clone H10 was used only in Western blot studies showing increased expression of Mcl-1. The manuscript would benefit from the performance of additional experiments including cell viability, oxidative stress, mitochondrial oxygen consumption, mitochondrial dynamics and mitophagy which would strengthen the results.

Response 6: Thank you for your suggestion. Indeed, with the results of 10H would strengthen the results and our hypothesis. Thus, we utilized 10H cells to perform cell viability and DNA content assay in Supplementary Figure 1, mitochondrial oxygen consumption in Supplementary Figure 3B, oxidative stress, mitochondrial dynamics and mitophagy in Supplementary Figure 5. We also added these experimental results to each "Results" section.

Question 7: Please correct the title in 3.4 section.

Response 7: Thank you for your kind suggestion. We corrected the title in 3.4 section as described below: Mcl-1 overexpression stabilized mitochondrial dynamics in cancer cells after IMQ treatment. (Page 8, Line 329)

Question 8: Please correct the legends in Fig. 4 and Fig. 5.

Response 8: Thank you for your kind suggestion. We have corrected the legends in Fig. 4 and Fig. 5 as described below:

Figure 4. Mcl-1 overexpression led to reduced IMQ-induced imbalance in mitochondrial dynamics in cancer cells. (A) Mcl-1 overexpression increased the mitochondrial length in IMQ-treated cancer cells. (B) Mcl-1 overexpression regulated the expression of fission- and fusion-related markers in IMQ-treated cancer cells. (A) After 50 µg/ml IMQ treatment for 24 hours, the cancer cells were stained with 100 nM MitoTracker Red CMXRos for 30 min. Mitochondrial morphology was observed by confocal microscopy, and the number of mitochondria in each morphological class was determined via CellSens Software (Olympus). Scale bars, 10 μm. (B) After 50 µg/ml IMQ treatment for 12 hours, cancer cells were harvested, and the expression of fission and fusion markers was measured via immunoblotting. The results were statistically analyzed by two-way ANOVA. P*<0.05, P**<0.01, and P***<0.001. (Page 10, Line 354-363)

Figure 5. Mcl-1 overexpression modulated IMQ-induced mitophagy in cancer cells. Mcl-1 overexpression not only decreased IMQ-induced colocalization of lysosomes and mitochondrial fragmentation (A) but also decreased IMQ-induced PINK1 expression in cancer cells (B). (A) Cells were preincubated with 100 nM MitoTracker Green FM for 30 min followed by 50 µg/ml IMQ treatment for 24 hours and then stained with 1 µM LysoTracker Red DND-99 for 30 min. The colocalization rates of mitochondria with lysosomes were monitored via confocal microscopy and analyzed with Olympus FV10-ASW V4.2 software (Olympus). Scale bars, 10 μm. (B) In preparation for immunoblotting, the cells were treated with 50 µg/ml IMQ for 12 hours, the levels of TOM20, PINK1 and β-actin were measured by immunoblotting. The results were statistically analyzed by two-way ANOVA. P*<0.05, P**<0.01, and P***<0.001.(Page 11, Line 388-397)

Question 9: The Conclusion section could be improved.

Response 9: Thanks for reviewer’s valuable comment. We have paraphrased and summarized the “Conclusion section” as below:

In this study, we found that Mcl-1 mitigated IMQ-induced oxidative stress and consequently protected cancer cells from death via IMQ. Furthermore, the decreased stability of Mcl-1 induced by IMQ is associated not only with the production of ROS but also with the maintenance of mitochondrial integrity in cancer cells. Mcl-1 overexpression enhanced mitochondrial fusion and stabilized the mitochondrial structure, providing cytoprotective effects in IMQ-treated cancer cells. Investigating the roles of Mcl-1 in mitochondria may lead to its designation as a potential target for IMQ-related cancer therapy. (Page 14, Line 512-518)

Question 10: The manuscript requires minor language correction

Response 10: This manuscript was edited for proper English language, grammar, punctuation, spelling, and overall style by one or more of the highly qualified native English-speaking editors at American Journal Experts. The corrections were highlighted in the revised manuscript. The editorial certificate also uploaded with the revised manuscript.

Reviewer 2 Report

Comments and Suggestions for Authors

Chang and coauthors submitted a manuscript titled “The protective effects of Mcl-1 on mitochondrial damage 2 and oxidative stress in imiquimod-induced tumor cell death.” The manuscript is a regular article.

In this work, the authors proposed an in-depth study on the role of MCL-1, a widely studied antiapoptotic protein, in ameliorating the function of mitochondria in cancer cells treated with Imiquimod, in terms of respiratory capacity, and the structural balance between fission and fusion. The authors adopted different experimental approaches, from the analysis of respiratory function, to immunoblotting and immunofluorescence with confocal microscopy, for the detection of mitochondrial function and mitophagy markers.

The results obtained are convincing and support the hypotheses put forward by the authors, in particular MCL1 acts in preserving the structure and functionality of mitochondria, and in particular the overexpression of MCL1 can determine greater resistance of cancer cells to treatment with the chemotherapeutic Imiquimod.

For authors I only have a few suggestions to make:

-Figures 1B and 6B, the control (BCC and AGS, black histograms), are not set to 100%; Are 12S and AGSIV compared to control (BCC and AGS)? Please specify this in the figure legend.

-Line 207, the WST test has never been described previously in the text. Please describe this text in the materials and methods section, or otherwise correct.

-Figure 2, it is unclear how fluorescence related to ROS production is measured and reported. The authors indicate the count of stained cells in the materials and methods, while the fluorescence value is reported in the graphs. Can the authors clarify this aspect and possibly better explain the methodology followed both in the materials and methods session and in the legend of the figure.

-In figure 4, the authors reported the measurements of mitochondria belonging to the three different classes (I-III). However, the authors did not indicate how this measurement was performed.

-Did the authors use both Mitotraker red and green? In the micrographs of Figure 4 and Supplementary Figure, mitochondria are colored as red fluorescence, and in green in others, please specify.

-In the summary Figure 6E, it is reported that MCL-1 would inhibit mitophagy, but the authors stated that overexpression of MCL-1 does not increase PINK1 expression. Should  this part of the figure be corrected? It is more important Tom20 or Pink1 in the characterization of mitophagy (Tom20 is more used as a mitochondria quantification marker)

Author Response

Response to Reviewer #2’ comments:

Thank you for your good comments and suggestions. According to your opinions, we have revised the manuscript, and the answers are below:

Question 1: Figures 1B and 6B, the control (BCC and AGS, black histograms), are not set to 100%; Are 12S and AGSIV compared to control (BCC and AGS)? Please specify this in the figure legend.

Response 1: We assessed cell viability by trypan blue staining assay in IMQ-treated cells. Thus, the cell viability in BCC and AGS control group is not 100% during three independent experiments. We correct this mistake in the Materials and Methods section (Page 4, Line 171), the Results section (Page 5, Line 250), the legend of Figure 1 (Page 6, Line 264), the legend of Figure 6 (Page 12, Line 421), and Supplementary Figure 1 (Page 15, Line 522-523).

Question 2: -Line 207, the WST test has never been described previously in the text. Please describe this text in the materials and methods section, or otherwise correct.

Response 2: Thank you for your kind suggestion. The cell viability in this paper were all conducted by trypan blue staining following the manufacturer’s protocol. We corrected this mistake and described as below: Through trypan blue staining assays, we consistently observed greater viability of the 10H, 12A and AGS IV cells than of control counterparts after IMQ treatment (Figure 1B and Supplementary Figure 1A). (Page 5, Line 249-252)

Question 3: Figure 2, it is unclear how fluorescence related to ROS production is measured and reported. The authors indicate the count of stained cells in the materials and methods, while the fluorescence value is reported in the graphs. Can the authors clarify this aspect and possibly better explain the methodology followed both in the materials and methods session and in the legend of the figure.

Response 3: Thank you for kindly suggestion. We utilized three fluorescence probes and flow cytometry to detect the oxidative stress (total ROS, cellular and mitochondrial superoxide) and analyze the differences between Mcl-1 overexpression cells and the control counterparts upon IMQ treatment. For easier to see the results from histograms, we transform each raw data (intensity of fluorescence) into graphs. We added overlay histograms in Figure 2 that would further help readers easier to see the results.

Question 4: In figure 4, the authors reported the measurements of mitochondria belonging to the three different classes (I-III). However, the authors did not indicate how this measurement was performed.

Response 4: Thank you for kindly suggestion. We added “Confocal imaging for assessing mitochondrial morphology” in section 2.8 as we previously described [1]. Mitochondria were categorized on the basis of their morphology via CellSens Software (Olympus). The mitochondria that were wholly fragmented or permeabilized were classified as fragmented mitochondria (Class 1), those with clear networks were labeled as tubular mitochondria (Class 3), and those with other shapes were categorized as intermediate mitochondria (Class 2). The proportions of mitochondria with these specific morphologies were calculated as a percentage of the total number of cells examined (100 cells per experiment). (Page 5, Line 221-227)

We added Confocal imaging for the colocalization of mitochondria and lysosomes in section 2.9 as we previously described [1].

Cancer cells were seeded in 6-well plates containing coverslips (2×105 cells/well) and allowed to reach 70–80% confluency. The cells were preincubated with 100 nM MitoTracker Green FM for 30 min followed by 50 µg/ml IMQ in 1 ml of medium for 24 hours and then stained with 1 µM LysoTracker Red DND-99 for 30 min. The colocalization rates of mitochondria with lysosomes were monitored via confocal microscopy and analyzed with Olympus FV10-ASW V4.2 software (Olympus). (Page 5, Line 229-235)

Question 5: Did the authors use both Mitotraker red and green? In the micrographs of Figure 4 and Supplementary Figure, mitochondria are colored as red fluorescence, and in green in others, please specify.

Response 5: Thank you for kindly suggestion. For assessing mitochondrial morphology, we stained cells with 100 nM MitoTracker Red CMXRos (Figure 4A, Figure 6D and Supplementary Figure 4) or 100 nM MitoTracker Green FM (Supplementary Figure 5B). The detailed information was addressed in section 2.8 (Page 5, Line 218-220). For colocalization of Mcl-1 and mitochondria (Supplementary Figure 2), we stained cells with 100 nM MitoTracker Red CMXRos for 30 min and then fixed with 1% formaldehyde in PBS at 4 °C overnight. The detailed information was addressed in section 2.7 (Page 4, Line 205-206). For assessing colocalization of mitochondria and lysosomes, we stained cells with 100 nM MitoTracker Green FM for 30 min (Figure 5A and Supplementary Figure 5C. The detailed information was addressed in section 2.9. (Page 5, Line 231-233)

Question 6: In the summary Figure 6E, it is reported that MCL-1 would inhibit mitophagy, but the authors stated that overexpression of MCL-1 does not increase PINK1 expression. Should this part of the figure be corrected? It is more important Tom20 or Pink1 in the characterization of mitophagy (Tom20 is more used as a mitochondria quantification marker)

Response 6: Thank you for kindly suggestion. We re-did the immunoblotting to compare the PINK expression between AGS control and AGS IV cells with or without IMQ treatment and confirm the result that overexpression of Mcl-1 did not increase PINK1 expression in 12A and AGS IV cells with IMQ treatment (Figure 5B). We found that Mcl-1 overexpression may reduce IMQ-induced mitophagy. After IMQ stimulation, the number of lysosomes increased and they colocalized with fragmented mitochondria in BCC and AGS cells. However, in 10H, 12A, and AGS IV cells, the number of colocalized mitochondria and lysosomes was significantly decreased. (Figure 5A and Supplementary Figure 5C). Notably, the expression of the mitophagy-related protein PINK1 did not significantly increase after IMQ treatment in 12A and AGS IV cells as compared with untreated 12A and AGS IV (Figure 5B). These findings suggested that IMQ does not trigger mitophagy in cancer cells overexpressing Mcl-1. This part was added in the Result section of 3.5. (Page 10, Line 381-383)

        TOM20 is essential for maintaining mitochondrial function and role in mitophagy is more about being a structural marker rather than a direct regulator [2]. PINK1 is crucial for initiating the mitophagy process, making it more directly involved in the characterization and regulation of mitophagy compared to TOM20. However, degradation of TOM20 also serves as a marker of mitophagy [3]. Both TOM20 and Pink1 play critical roles in mitophagy. In this study, we not only focused on the regulation and initiation of mitophagy but also interested in identifying or tracking mitochondria. That is why we chose both TOM20 and Pink1 as mitophagy markers.

  1. Chuang, K.C.; Chang, C.R.; Chang, S.H.; Huang, S.W.; Chuang, S.M.; Li, Z.Y.; Wang, S.T.; Kao, J.K.; Chen, Y.J.; Shieh, J.J. Imiquimod-induced ROS production disrupts the balance of mitochondrial dynamics and increases mitophagy in skin cancer cells. Journal of dermatological science 2020, 98, 152-162, doi:10.1016/j.jdermsci.2020.03.009.
  2. Ding, W.X.; Yin, X.M. Mitophagy: mechanisms, pathophysiological roles, and analysis. Biological chemistry 2012, 393, 547-564, doi:10.1515/hsz-2012-0119.
  3. Chan, N.C.; Salazar, A.M.; Pham, A.H.; Sweredoski, M.J.; Kolawa, N.J.; Graham, R.L.; Hess, S.; Chan, D.C. Broad activation of the ubiquitin-proteasome system by Parkin is critical for mitophagy. Human molecular genetics 2011, 20, 1726-1737, doi:10.1093/hmg/ddr048.

Reviewer 3 Report

Comments and Suggestions for Authors

The article by Dr. Shieh and the group elaborates on the role of Mcl-1 on mitochondria damage and oxidative stress and mechanistically explains how that affects cancer cell death. This is a very well-written manuscript with hypothetically well-planned experiments. This article also sheds light on therapeutic implications. However, a few things must be addressed before it is ready for acceptance. They are as follows:

1. It's been known that NRF2/ antioxidant pathways play a definitive role in oxidative stress (PMID: 23294312). It has also been shown how NRF2 plays a role in creating chemoresistance (PMID: 31911550). It will be worthwhile to shed some light on how MCL1 might regulate NRF2-mediated oxidative stress or vice versa. Authors should add a few lines discussing this aspect in the discussion part as one of the future aspects of this study by adding relevant references. 

2. Fig 4 a, graphs can be put below/downside of the IF pictures so that the figure looks better. All the figures must be reorganized, and the text inside the figures should be enlarged to be readable.

3. Fig 4b, p DRP1- please mention the phosphorylation site next to the western blot figure. 

Author Response

Response to Reviewer #3’ comments:

Thank you for your good comments and suggestions. According to your opinions, we have revised the manuscript, and the answers are below:

Question 1: It's been known that NRF2/ antioxidant pathways play a definitive role in oxidative stress (PMID: 23294312). It has also been shown how NRF2 plays a role in creating chemoresistance (PMID: 31911550). It will be worthwhile to shed some light on how MCL1 might regulate NRF2-mediated oxidative stress or vice versa. Authors should add a few lines discussing this aspect in the discussion part as one of the future aspects of this study by adding relevant references. 

Response 1: Thanks for reviewer’s valuable comment. The interplay between Mcl-1 and Nrf2 in IMQ-induced cell death and oxidative stress is an important issue. We addressed the possible role of Mcl-1 in Nrf2 regulation as described below: Nuclear factor erythroid 2–related factor 2 (Nrf2) not only plays a definitive role in the regulation of oxidative stress but also contributes to chemoresistance, which Mcl-1 also promotes in multiple types of cancer [1-5]. In this study, we found that IMQ-induced production of ROS and mitochondrial superoxide was significantly abolished in Mcl-1-overexpressing BCC and AGS cancer cells (Figure 2 and Supplementary Figure 5A). Nrf2 directly inhibits apoptosis by upregulating the expression of BCL-2 and BCL-XL [6]. However, the upregulation of BCL-2 enhances Nrf2 activity[7]. It is possible that Mcl-1 may enhance Nrf2 activation to protect cancer cells from IMQ-induced oxidative stress. The crosstalk between Mcl-1 and Nrf2 in IMQ-induced cell death and oxidative stress is a significant concern and unknown. We will further investigate these issues. These parts were addressed in the second paragraph of the Discussion section. (Page 13, Line 441-450)

Question 2: Fig 4 a, graphs can be put below/downside of the IF pictures so that the figure looks better. All the figures must be reorganized, and the text inside the figures should be enlarged to be readable.

Response 2: Thank you for your kind suggestion. We not only reorganized and enlarged Figure 4A, but also adjusted other figures to ensure they are readable. 

Question 3: Fig 4b, p DRP1- please mention the phosphorylation site next to the western blot figure. 

Response 3: Thank you for kindly suggestion. The phosphorylation site for p-DRP1 is Ser616, thus we labeled p-DRP1 as p-DRP1 (Ser616) in Figure 4B. (Page 10)

  1. Fu, D.; Pfannenstiel, L.; Demelash, A.; Phoon, Y.P.; Mayell, C.; Cabrera, C.; Liu, C.; Zhao, J.; Dermawan, J.; Patil, D.; et al. MCL1 nuclear translocation induces chemoresistance in colorectal carcinoma. Cell death & disease 2022, 13, 63, doi:10.1038/s41419-021-04334-y.
  2. Bosc, C.; Selak, M.A.; Sarry, J.E. Resistance Is Futile: Targeting Mitochondrial Energetics and Metabolism to Overcome Drug Resistance in Cancer Treatment. Cell metabolism 2017, 26, 705-707, doi:10.1016/j.cmet.2017.10.013.
  3. Liu, A.S.; Yu, H.Y.; Yang, Y.L.; Xue, F.Y.; Chen, X.; Zhang, Y.; Zhou, Z.Y.; Zhang, B.; Li, L.; Sun, C.Z.; et al. A Chemotherapy-Driven Increase in Mcl-1 Mediates the Effect of miR-375 on Cisplatin Resistance in Osteosarcoma Cells. OncoTargets and therapy 2019, 12, 11667-11677, doi:10.2147/OTT.S231125.
  4. Ma, Q. Role of nrf2 in oxidative stress and toxicity. Annual review of pharmacology and toxicology 2013, 53, 401-426, doi:10.1146/annurev-pharmtox-011112-140320.
  5. Mukhopadhyay, S.; Goswami, D.; Adiseshaiah, P.P.; Burgan, W.; Yi, M.; Guerin, T.M.; Kozlov, S.V.; Nissley, D.V.; McCormick, F. Undermining Glutaminolysis Bolsters Chemotherapy While NRF2 Promotes Chemoresistance in KRAS-Driven Pancreatic Cancers. Cancer research 2020, 80, 1630-1643, doi:10.1158/0008-5472.CAN-19-1363.
  6. Cha, H.Y.; Lee, B.S.; Chang, J.W.; Park, J.K.; Han, J.H.; Kim, Y.S.; Shin, Y.S.; Byeon, H.K.; Kim, C.H. Downregulation of Nrf2 by the combination of TRAIL and Valproic acid induces apoptotic cell death of TRAIL-resistant papillary thyroid cancer cells via suppression of Bcl-xL. Cancer letters 2016, 372, 65-74, doi:10.1016/j.canlet.2015.12.016.
  7. Ruefli-Brasse, A.; Reed, J.C. Therapeutics targeting Bcl-2 in hematological malignancies. The Biochemical journal 2017, 474, 3643-3657, doi:10.1042/BCJ20170080.

Round 2

Reviewer 1 Report

Comments and Suggestions for Authors

ok

Reviewer 3 Report

Comments and Suggestions for Authors

All concerns addressed and ready for acceptance.